# Potential enablers for the implementation of multiple family group therapy intervention in the lower Manya Krobo District, Ghana: Perspectives of multiple stakeholders

Dorothy Serwaa Boakye[1], Samuel Adjorlolo[2]*

1 Department of Health Administration and Education, University of Education, Winneba, Ghana,
2 Department of Mental Health Nursing, School of Nursing and Midwifery, University of Ghana, Accra, Ghana

* sadjorlolo@ug.edu.gh

## Abstract

The Multiple Family Group Therapy (MFGT) has demonstrated effectiveness in addressing behavioral problems and mental health disorders among adolescents in high-income settings. Limited evidence exists on whether the MFGT can be implemented successfully in Ghana and other SSA for adolescents living with HIV (ALHIV), given the contextual differences between Ghana and high-income countries. This study aimed to identify factors and processes from multiple stakeholders that can support a successful implementation of the MFGT intervention for ALHIV and their families in the Lower Manya Krobo District of Ghana. A qualitative exploratory study was conducted at Atua Government Hospital. Data were collected through focus group discussions with ALHIV (n = 12), caregivers (n = 13), and health professionals (n = 5), supplemented by three in-depth interviews (with ALHIV). Contextual thematic analysis was employed to identify patterns and themes within the data, with attention to rigor through triangulation, member checking, and reflexivity. Six key enablers for MFGT implementation emerged: (1) existing support systems and infrastructure, including established HIV programs and dedicated adolescent clinic days; (2) willingness of health workers to participate and sustain the intervention; (3) practical implementation considerations including scheduling preferences and privacy concerns; (4) participation support needs such as material and logistical support and preference for diverse educational approaches (5) content preferences emphasizing comprehensive topics beyond HIV management and interactive learning approaches. Five key enablers emerged for successful MFGT implementation. Implementation strategies should integrate MFGT into existing adolescent clinic days, provide transportation assistance and reminder systems, accommodate Sunday scheduling preferences, train diverse facilitators, including peer mentors, and incorporate spiritual elements while maintaining flexible group compositions based on discussion topics.

**Data availability statement:** All data are in the manuscript and supporting information files.

**Funding:** The study was made possible (in-part) by the Preparing Outstanding Social Science Investigators to Benefit Lives and Environments in Africa Initiative (POSSIBLE-Africa) (Grant NO# POS-24-07 to S.A), an initiative of the Science for Africa Foundation (SFA Foundation) enabled by the support of Carnegie Corporation of New York. No additional external funding was received for this study.

**Competing interests:** The authors have declared that no competing interests exist.

## Introduction

The Multiple Family Group Therapy (MFGT) has emerged as an effective psychosocial intervention for addressing the complex needs of individuals and families affected by chronic health conditions [1]. MFGT brings together multiple families to share experiences, develop coping strategies, and provide mutual support under professional guidance [2]. Previous studies have consistently demonstrated the efficacy of MFGT across diverse populations and health conditions, with studies showing significant improvements in treatment adherence, family functioning, and psychological well-being [3,4].

Research from high-income countries has identified several key enablers for successful MFGT implementation. Organizational support, including dedicated time, space, and resources for MFGT sessions, has been consistently identified as critical for implementation success [5]. Professional training and supervision for facilitators have also been highlighted as essential enablers, with studies showing that implementation fidelity is significantly higher when facilitators receive specialized training and ongoing supervision [6]. Integration with existing care systems represents another important enabler, with successful MFGT programs often embedded within established treatment pathways rather than operating as standalone interventions [7]. Additionally, adaptability of the MFGT model to different clinical populations and settings has facilitated wider implementation, with flexible protocols allowing practitioners to maintain core therapeutic principles while adjusting content to meet specific population needs [8]. Stakeholder buy-in, particularly from organizational leadership and frontline providers, has been identified as a crucial enabler of successful implementation in high-income settings [9].

In resource-rich countries, MFGT has shown promising results in addressing adolescents' behavioral problems and mental disorders such as depression and anxiety. Studies have demonstrated that MFGT can be particularly effective for reducing internalizing behaviors in adolescents with disruptive behavior disorders [10,11]. Family-based interventions for adolescent depression have been found to have small but significant effects compared to active control conditions without family involvement [12], suggesting that family participation can enhance treatment outcomes.

For adolescents living with HIV (ALHIV), psychosocial support is particularly crucial as they navigate both the normative challenges of adolescence and the additional burdens of HIV management, stigma, and disclosure concerns [13,14]. Despite advances in medical treatment, ALHIV continue to face significant challenges in adherence, mental health, and social integration [15]. In sub-Saharan Africa, where the burden of HIV among adolescents remains disproportionately high, these challenges are often exacerbated by resource constraints, cultural factors, and persistent stigma [16]. Ghana's Lower Manya Krobo District represents a microcosm of these challenges, with high HIV prevalence and limited psychosocial support systems for ALHIV [17,18].

The implementation of MFGT in resource-limited settings like Ghana faces distinct contextual challenges that require careful consideration. HIV-related stigma remains

pervasive in many African communities and can significantly impact willingness to participate in group-based interventions where HIV status might be disclosed to others [19]. Resource constraints present another significant contextual challenge, with many healthcare facilities in sub-Saharan Africa facing shortages of trained mental health professionals, limited physical space for group interventions, and competing demands for available resources [20]. Cultural conceptualizations of mental health, family dynamics, and help-seeking also differ significantly between Western contexts where MFGT originated and African settings. In many African communities, collectivist cultural values, spirituality, and traditional healing practices play important roles in addressing psychological challenges [21]. Furthermore, the socioeconomic realities of many families in resource-limited settings can impact participation, with transportation costs, lost income from attending sessions, and competing survival priorities potentially limiting engagement with MFGT [22].

While MFGT has shown promise in high-income settings [1,2,23], there remains a significant gap in our understanding regarding the local and contextual factors in Ghana that can support its implementation. The existing literature predominantly focuses on intervention outcomes rather than implementation processes, leaving critical questions about contextual adaptations, sustainability mechanisms, and stakeholder perspectives largely unexplored. Moreover, few studies have specifically examined the implementation of family-based interventions for ALHIV in sub-Saharan Africa [24,25], despite the recognized importance of family support in HIV management.

Given these contextual complexities, the perspectives of multiple stakeholders become critical for successful MFGT implementation. Health workers possess invaluable insights about existing health systems, resource limitations, and potential integration points that could facilitate implementation [26]. ALHIV perspectives are essential for understanding potential barriers to participation, preferences regarding intervention format and content, and concerns about confidentiality and stigma. As the primary recipients of the intervention, their engagement in the implementation process is vital for ensuring that MFGT addresses their actual rather than presumed needs [27]. Caregivers' perspectives provide crucial information about family dynamics, cultural norms regarding parenting and adolescent development, and practical considerations that might affect family participation. Their insights can help identify culturally appropriate methods for engaging families and addressing potential resistance [28].

This study addresses these critical gaps by exploring the enablers for successful implementation of an adapted MFGT intervention in the Lower Manya Krobo District of Ghana. By examining the perspectives of multiple stakeholders—health workers, caregivers, and ALHIV—the research provides valuable insights into the practical, cultural, and systemic factors that could facilitate effective implementation. Understanding these enablers is essential for developing culturally responsive, sustainable interventions that can be integrated into existing healthcare systems. The findings from this study will not only inform the implementation of MFGT in this specific context but also contribute to the broader understanding of how evidence-based family interventions can be successfully adapted and implemented in diverse settings across sub-Saharan Africa, ultimately improving outcomes for ALHIV and their families.

## Theoretical grounding of the study

This study was guided by the Consolidated Framework for Implementation Research (CFIR), a comprehensive meta-theoretical framework that synthesizes implementation theories to understand the factors influencing the effectiveness of intervention implementation [29,30]. CFIR's five domains- Inner setting, Outer setting, Intervention characteristics, Individual characteristics and Implementation process- provide a systematic lens for examining implementation enablers across multiple levels of analysis [31], making it particularly suited for exploring stakeholder perspectives on MFGT implementation in complex healthcare settings.

CFIR's multi-stakeholder approach aligns with this study's inclusion of healthcare professionals, caregivers, and adolescents living with HIV, recognizing that implementation success depends on understanding perspectives across the healthcare ecosystem. The framework's emphasis on contextual adaptation is particularly relevant for implementing evidence-based interventions like MFGT in resource-limited settings where cultural, organizational, and economic factors

may differ significantly from high-income contexts where the intervention was originally developed. By applying CFIR domains to guide data collection and analysis, this study systematically explores enablers across all levels that influence implementation, providing a theoretically grounded foundation for understanding how MFGT can be successfully adapted and sustained in Ghana's healthcare system.

## Methods

### Ethics statement

The study received ethical approval from the Ghana Health Service Ethics Review Committee (GHS-ERC:004/07/24) on July 15, 2024. No waiver was required as full ethics review was completed. Eastern Regional Health Directorate and Hospital administration provided institutional permission. Several measures were implemented to ensure ethical conduct of the research. For ALHIV under 18 years, informed consent was obtained from caregivers and assent from the adolescents themselves. Adult participants provided written informed consent. All consent and assent processes were conducted in participants' preferred language.

Confidentiality was maintained through the use of ID numbers in all documentation and secure storage of data. Given the sensitive nature of HIV status and the potential for psychological distress during discussions, a clinical psychologist was available on-site during all data collection sessions to provide support if needed. Participants were informed of their right to withdraw from the study at any time without consequences for their healthcare or other services. Special attention was paid to the unique vulnerabilities of ALHIV, with age-appropriate communication strategies employed during recruitment and data collection. Transportation stipends were provided to all participants to prevent financial burden, and refreshments were served during FGDs to create a comfortable atmosphere.

### Study setting

The study was conducted at Atua Government Hospital, located in the Lower Manya Krobo District in the Eastern Region of Ghana. This facility was purposively selected because it serves as one of the primary healthcare centers in the district providing comprehensive HIV services, including specialized care for adolescents living with HIV (ALHIV). The hospital has established adolescent-specific clinic days and operates several HIV support programs, making it an ideal setting to explore implementation enablers for a Multiple Family Group Therapy (MFGT) intervention. The facility's catchment area encompasses both urban and rural communities with varying socioeconomic profiles, providing access to diverse perspectives on implementation considerations.

### Study design

A qualitative exploratory design was employed to gain in-depth understanding of the perspectives of multiple stakeholders regarding enablers for implementing MFGT in the Lower Manya Krobo District. The exploratory nature of the design was particularly appropriate given the limited previous research on MFGT implementation in sub-Saharan African settings, allowing for the identification of novel, culturally-specific implementation enablers that might not be captured in existing frameworks developed in high-income countries.

### Study participants and sampling

Three key stakeholder groups were included in the study: adolescents living with HIV (ALHIV), caregivers of ALHIV, and healthcare professionals working with ALHIV. A purposive sampling strategy was used to recruit participants who could provide information-rich perspectives on the research questions. Inclusion criteria for ALHIV were: (1) aged 12–22 years, (2) aware of their HIV status, (3) receiving care at Atua Government Hospital, and (4) willing and able to participate with caregiver consent and their own assent/consent. Caregivers were eligible if they were the primary caregiver for an ALHIV

 

enrolled in care at the facility and willing to participate. Healthcare professionals were included if they had at least one year of experience working with ALHIV at the facility.

The number of participants was guided by the principle of theoretical saturation, ensuring that data collection continued until no new themes emerged across stakeholder groups. Additionally, the sample size balanced practical and logistical considerations while maintaining sufficient diversity across adolescents living with HIV, caregivers, and healthcare professionals to enable robust triangulation of perspectives and a comprehensive understanding of implementation enablers within the healthcare ecosystem.

## Design of instruments

Two types of data collection instruments were developed for the study: focus group discussion (FGD) guides and in-depth interview guides. The development of these instruments was informed by relevant literature on implementation science frameworks, particularly the Consolidated Framework for Implementation Research (CFIR), and adapted to address the specific context of MFGT implementation for ALHIV in Ghana. Both instruments were organized around key domains including perceptions of existing support systems, willingness to participate, cultural and community factors, practical implementation considerations, support needs, and content preferences.

The FGD guides included open-ended questions designed to stimulate group discussion and encourage interaction between participants, with appropriate probes and prompts to elicit deeper insights. Separate guides were developed for each stakeholder group to address their unique perspectives (S1Text, S2 Text, S3 Text). The interview guides contained more detailed questions to explore individual experiences and perspectives in greater depth. All instruments were reviewed by experts in implementation science and adolescent HIV care, then pilot-tested with 2 representatives from each stakeholder group not participating in the main study. Feedback from the pilot testing was incorporated to improve clarity, cultural appropriateness, and relevance of the questions.

## Data collection procedure

Participant recruitment took place from January 15–29, 2025, following ethics approval. Recruitment allowed participants adequate time to consider participation, and data collection commenced on February 1, 2025, and continued through to March 30, 2025, after all participants provided informed consent or assent. Focus group discussions were the primary data collection method, supplemented by individual interviews to capture more sensitive or detailed perspectives. For ALHIV, two FGDs were conducted with 4–6 participants per group, segregated by gender to encourage open discussion of potentially sensitive topics. For caregivers, two FGDs were conducted with 6 and 7 participants per group. Healthcare professionals participated in one FGD with all 5 professionals.

Additionally, three in-depth interviews were conducted with ALHIV to gather more detailed perspectives. These individuals were selected based on their particularly rich experiences or unique viewpoints identified during recruitment. Data collection occurred through two distinct modalities.. Individual interviews were conducted online via telephone or Google Meet to ensure participants' privacy, convenience, and minimal disruption to their schedules. To address potential network issues, interviews were rescheduled as needed, and telephone calls were offered as an alternative to videoconferencing. This flexible approach maintained data quality, participant engagement, and confidentiality. FGD were conducted face-to-face.

For FGD, the research team prioritized participant privacy by utilizing dedicated private spaces within Atua Government Hospital. These sessions were led by the 1st author (a female lecturer and PhD candidate with expertise in qualitative research and HIV) with crucial support from a research assistant who performed dual functions: providing immediate language interpretation between English and local languages (primarily Dangme and Twi) as needed, while simultaneously documenting session notes. This bilingual support ensured all participants could fully engage regardless of their language preference.

FGDs lasted approximately 90–120 minutes, while interviews lasted 60–90 minutes depending on participant responses and depth of discussion. With participants' permission, all sessions were audio-recorded and supplemented with field notes documenting non-verbal communications and contextual observations. After each session, a debriefing meeting was held among the research team to discuss emerging themes and refine the approach for subsequent sessions.

Challenges encountered during the study, included scheduling conflicts, occasional network interruptions during online interviews, language translation complexities, and ensuring participant comfort and confidentiality. To mitigate these challenges, we implemented several strategies: interviews were rescheduled as needed, telephone calls were offered as an alternative to videoconferencing, bilingual research assistants assisted with accurate translation and interpretation, and private, comfortable settings were arranged for both FGDs and individual interviews to maintain confidentiality and participant ease.

### Research team, training, and standardization

The research team comprised two main members—DSB, a Ghanaian female PhD candidate and lecturer with four years' experience in qualitative health research and HIV/AIDS, and SA, a Ghanaian male professor of Mental Health Nursing with over six years' experience in adolescent care and qualitative research methods—supplemented by two Ghanaian research assistants and one health worker who assisted with participant recruitment and eligibility screening. Both lead researchers are affiliated with public universities in Ghana, bringing institutional credibility but also potential power dynamics related to academic status. DSB's background as a health education specialist and SA's clinical nursing background shaped their theoretical orientations toward health systems and psychosocial wellbeing. The research assistants, who were younger and from the local community, helped bridge the cultural and generational gaps between the academic researchers and participants, particularly those with ALHIV.

Before data collection, the team underwent a one-day intensive training to ensure consistency in applying research instruments and interview techniques. Training covered qualitative interviewing principles, instrument review, cultural sensitivity specific to ALHIV and their families, ethical considerations, and data management protocols. The research assistants received additional language-specific training to ensure accurate interpretation. Pilot interviews and debriefing sessions were conducted to refine techniques, instruments, and standardize procedures across the team.

### Data analysis

A contextual thematic analysis approach was employed to analyze the data, allowing for a systematic examination of explicit and implicit meanings within the dataset. The analysis process began with verbatim transcription of all audio recordings. Transcripts in local languages were translated into English by bilingual research team members, with back-translation of selected segments to ensure accuracy.

The analysis followed Braun and Clark [32] six key steps: (1) familiarization with the data through repeated reading of transcripts and review of field notes; (2) generation of initial codes through line-by-line coding of transcripts; (3) searching for themes by collating codes into potential themes; (4) reviewing themes in relation to the coded extracts and entire dataset; (5) defining and naming themes; and (6) producing the report with selected vivid, compelling extracts.

QDA Miner software was used to facilitate data management and coding. The initial coding framework was developed based on the research questions and interview guides, then refined inductively as analysis progressed. To enhance theoretical rigor, the CFIR framework was applied to organize and interpret codes and emerging themes across its five domains, ensuring that findings were mapped systematically to relevant constructs at multiple levels of implementation. The two authors independently coded a subset of transcripts to establish coding consistency, with discrepancies resolved through discussion. The analysis was conducted iteratively, moving between data collection and analysis to allow emerging themes to inform subsequent data collection. Themes were developed within and across stakeholder groups to identify both common and divergent perspectives.

PLOS Global Public Health

### Rigor and reflexivity

The study ensured trustworthiness through systematic application of Lincoln and Guba's [33] four criteria—credibility, transferability, dependability, and confirmability—aligned with Consolidated Criteria for Reporting Qualitative Research (COREQ) guidelines [34]. Credibility was established via triangulation across multiple stakeholder groups (healthcare professionals, caregivers, and ALHIV) and methods (FGDs and individual interviews), member checking with six participants, prolonged engagement over four weeks, and peer debriefing sessions. Transferability was enhanced through thick description of the study context, participant characteristics, and detailed presentation of findings with rich quotes, allowing readers to assess applicability to similar settings. Dependability was supported by a comprehensive audit trail documenting methodological decisions, coding processes, and inter-coder reliability checks.

Confirmability was addressed through researcher triangulation, systematic reflexivity practices, and maintenance of reflexive journals to track biases and assumptions. Constant comparison methods and negative case analysis ensured findings were grounded in participant perspectives rather than researcher preconceptions. Overall, adherence to COREQ standards [34], including transparent reporting of the research team, study design, and analytical processes, strengthens the rigor of the study and allows readers to evaluate the credibility, dependability, and transferability of the findings.

Reflexivity was maintained through regular team discussions, reflexive journaling, and peer debriefing about how researchers' positionality influenced data collection and interpretation. The lead researchers—DSB (Ghanaian female lecturer, four years' HIV/AIDS research experience) and SA (Ghanaian male professor, mental health nursing background)—were affiliated with public universities, positioning them as "educated outsiders" with potential power imbalances relative to participants. However, their cultural familiarity as Ghanaian researchers fluent in local languages (Twi, basic Dangme) and understanding of community norms facilitated rapport-building. To mitigate power imbalances with vulnerable ALHIV, strategies included: emphasizing study independence from clinical care, conducting private interviews without authority figures, utilizing younger bilingual research assistants as bridges, and positioning participants as experts. The researchers acknowledged their implementation science training could privilege clinical effectiveness over community priorities; initial interview guides using Western therapeutic terminology were revised after pilot testing revealed participants' holistic conceptualizations incorporating spiritual and communal dimensions. The team's insider-outsider positioning—academically trained researchers studying their own cultural context—required constant critical examination of assumptions while avoiding presumptions of community homogeneity.

### Findings

**Sociodemographic characteristics of participants.** Most adolescents living with HIV (ALHIV) were aged 15–19 years (58.3%), with a predominance of females (83.3%). The majority were students (83.3%) and had attained at least junior high school education. Awareness of HIV diagnosis varied, with 41.6% having known their status for 1–5 years. A third lived with grandparents, and caregiver's marital status was mostly married (33.3%) or widowed (25%). The healthcare worker sample included both males and females, predominantly nurses, with a wide age distribution. Most had worked in the HIV unit for more than one year and had received some form of mental health (80%) or psychosocial support training (60%). Among caregivers, the majority were female (92.3%) and Christian (100%), with ages ranging from 25 to over 60 years. Most had at least junior high school education and were primarily mothers (38.5%) or other close relatives. Nearly 40% had been caregiving for over 10 years, while a significant proportion (61.5%) reported not receiving any external support. (see Table 1,Table 2 and Table 3).

### Main findings

**Themes and subthemes.** The thematic analysis of the FGD and interview transcript revealed several interconnected themes and subthemes related to the enablers for implementing an adapted MFGT intervention in the Lower Manya

**Table 1. Sociodemographic characteristics of ALHIV.**

| Variable | Category | Frequency | Percentage |
|---|---|---|---|
| Age | 10–14 | 3 | 25.0 |
| | 15-19 | 7 | 58.3 |
| | 20-24 | 2 | 16.7 |
| Gender | Male | 3 | 25.0 |
| | Female | 9 | 75.0 |
| Educational level | Primary | 1 | 8.3 |
| | Junior High school | 4 | 33.3 |
| | Senior High School | 5 | 41.6 |
| | Completed JHS | 2 | 16.7 |
| Occupation | Student | 10 | 83.3 |
| | Learning trade | 2 | 16.7 |
| Years since HIV Diagnosis | < 1 yr | 3 | 25.0 |
| | 1-5yrs | 5 | 41.6 |
| | >5yrs | 4 | 33.3 |
| Lives with | Parents | 2 | 16.7 |
| | Mother only | 3 | 25.0 |
| | Grandparents | 4 | 33.3 |
| | Siblings | 3 | 25.0 |
| Guardian Marital Status | Single | 3 | 25.5 |
| | Married | 4 | 33.3 |
| | Widowed | 3 | 25.5 |
| | Divorced | 2 | 16.7 |

Krobo District. These themes represent the perspectives of Health workers, caregivers and ALHIV and highlight critical considerations for successful program implementation. **(s**ee Table 4)

**Comparative synthesis of stakeholder perspectives on the potential enablers MFGT implementation**

This data presents a comparative synthesis of perspectives across the three stakeholder groups (caregivers, ALHIV, and health workers) regarding key implementation enablers. The table systematically maps areas of consensus and divergence, with consensus levels indicating the degree of alignment: high consensus suggests strong agreement across all groups, moderate consensus indicates general alignment with some nuanced differences, and low consensus highlights areas of significant divergence requiring careful attention in implementation planning (Table 5).

**CFIR Domain: inner setting**

**Theme 1: existing support systems & infrastructure**

**Subtheme 1.1: existing HIV support programs:.** Participants highlighted formal support systems already in place that specifically target people living with HIV, indicating potential integration points for the new therapy model. These programs have established protocols, dedicated staff, and existing participant networks that could facilitate the introduction of family-focused interventions.

*"We have what we call the Community Refill Program... creates room for groups of 10... we are to invite them. And then we have discussions, topics like disclosure to partners, sometimes on viral load, sometimes even on adherence and treatment support..."* (Health professional 3)

**Table 2. Sociodemographic characteristics of Caregivers.**

| Variable | Category | Frequency | Percentage |
|---|---|---|---|
| Age | 25-30 | 1 | 7.7 |
| | 31-40 | 5 | 38.5 |
| | 41-50 | 2 | 15.4 |
| | 51-60 | 1 | 7.7 |
| | >60 | 4 | 30.8 |
| Gender | Male | 1 | 7.7 |
| | Female | 12 | 92.3 |
| Educational level | No formal education | 1 | 7.7 |
| | Junior High School | 6 | 46.2 |
| | Senior High School | 2 | 15.4 |
| | Ordinary Level | 4 | 30.8 |
| Religion | Christian | 13 | 100 |
| Marital status | Single | 3 | 23.1 |
| | Married | 4 | 30.8 |
| | Widowed | 4 | 30.8 |
| | Divorced | 2 | **15.4** |
| Relationship to Adolescent | Mother | 5 | 38.5 |
| | Sibling | 2 | 15.4 |
| | Grandmother | 3 | 23.1 |
| | Guardian | 3 | 23.1 |
| Duration of caregiving | ≤1 year | 4 | 30.8 |
| | >1–10 years | 3 | 23.1 |
| | >10 years | 5 | 38.5 |
| | Not clearly specified | 1 | 7.7 |
| Receives support | Yes | 5 | 38.5 |
| | No | 8 | 61.5 |

This quote demonstrates that group-based interventions are already operationalized within the health system, suggesting that the structural framework for gathering people living with HIV already exists. The existing practice of bringing together approximately ten individuals suggests an established optimal group size that could inform the MFGT design.

Another participant noted:

*"Programs like the community adolescent treatment supporters exist already in the facility and can be leveraged. This program brings together trained adolescent mentors with optimal adherence and sustained decrease in viral load to provide adherence and psychosocial support to other adolescents living with HIV."* (Health professional, 2)

This quote is particularly significant as it highlights a peer-support model already functioning in the facility. The community adolescent treatment supporters program represents an existing resource of trained adolescent mentors who could potentially serve as facilitators or champions for the new family therapy intervention.

**Subtheme 1.2: Available human resources.** The availability of committed healthcare workers and support staff emerged as a critical foundational element upon which the intervention could be built. While material resources and funding might present challenges, participants emphasized that personnel were willing and prepared to implement new supportive interventions.

**Table 3. Sociodemographic of Healthcare workers.**

| Variable | Category | Frequency | Percentage |
|---|---|---|---|
| Age | 30-39 | 2 | 40 |
| | 40-49 | 1 | 20 |
| | 50+ | 2 | 40 |
| Gender | Male | 2 | 40 |
| | Female | 3 | 60 |
| Professional role | Nurse | 4 | 80 |
| | Mentor mother | 1 | 20 |
| Years in HIV unit | <1 year | 1 | 20 |
| | 1-5 years | 2 | 40 |
| | >5 years | 2 | 40 |
| Educational level | JSS | 1 | 20 |
| | Diploma | 1 | 20 |
| | Degree | 1 | 20 |
| | Not specified | 2 | 40 |
| Facility type | District | 5 | 100 |
| Mental Health Training | Yes | 4 | 80 |
| | No | 1 | 20 |
| Psychosocial support training | Yes | 3 | 60 |
| | No | 2 | 40 |

*"For now I can only speak of human resources. The human resource...for that we are available, and we are always ready to help it going, yes."* (Health professional 5)

This straightforward but powerful statement indicates a strong commitment from the healthcare staff to support the implementation of the MFGT program. The participants' emphasis on human resources suggests this might be perceived as the most reliable existing asset compared to other potential resources

**Subtheme 1.3: Dedicated clinic days for adolescents.** The focus group discussions revealed that the healthcare facility has already established specialized clinic days dedicated to adolescents living with HIV. This organizational infrastructure represents a significant enabling factor for implementing MFGT, as it provides an established timeframe and space where adolescents are already congregating for care.

*"The facility and the unit decided to now set specific days aside for these adolescents... we have Wednesdays. So, Wednesdays are just for adolescents and it's a way of preventing even further, I mean psychological problems and emotional problems along their life."* (Health professional 3)

This quote highlights several important aspects of the existing support infrastructure. First, it demonstrates institutional commitment to adolescent care through the deliberate allocation of specific clinic days. The participants explicitly connect these dedicated clinic days to psychological and emotional wellbeing, showing an existing awareness of mental health needs that aligns with the goals of MFGT.

## CFIR domain: individual characteristics

### Theme 2: willingness of health workers to participate

**Subtheme 2.1: readiness for training.** The focus group discussions revealed a high level of readiness among health workers to receive training for implementing the MFGT intervention. This willingness to engage in capacity building

**Table 4. Thematic mapping to CFIR domains.**

| CFIR Domain | Themes | Subthemes | Codes | Representative quote |
|---|---|---|---|---|
| INNER SETTING | Theme 1: Existing Support Systems & Infrastructure | 1.1 Existing HIV support programs | • Community Refill Program<br>• Established protocols<br>• Dedicated staff networks | "We have what we call the Community Refill Program... creates room for groups of 10... we have discussions, topics like disclosure to partners" (HP3) |
| | | 1.2 Available human resources | • Committed healthcare workers<br>• Willingness to implement<br>• Staff availability | "For now I can only speak of human resources... we are available, and we are always ready to help it going (HP5) |
| | | 1.3 Dedicated clinic days | • Wednesday- adolescent clinics<br>• Specialized scheduling<br>• Institutional commitment | "The facility... set specific days aside for these adolescents... Wednesdays are just for adolescents" (HP3) |
| INDIVIDUAL CHARACTERISTICS | Theme 2: Willingness of Health Workers to Participate | 2.1 Readiness for training | • Workshop attendance willingness<br>• Professional development openness<br>• Staff availability for training | "Once you organize a workshop for us, it will be okay. And we, the staff, are readily available" (HP2) |
| | | 2.2 Commitment to sustainability | • Knowledge transfer emphasis<br>• Long-term continuation plans<br>• Staff ownership development | "Once you've trained us, you've gained that knowledge... we'll continue with the intervention once the program has even ended" (HP1) |
| OUTER SETTING | Theme 3: Practical Implementation Considerations | 3.1 Scheduling and timing | • Sunday after church preference<br>• Religious activity accommodation<br>• Community rhythm alignment | "We prefer Sunday after church" (Multiple participants) |
| | | 3.2 Location and environment | • Privacy concerns<br>• Hospital vs. community settings<br>• Stigma considerations | "The environment we would meet shouldn't be an open place where other people would hear us" (ALHIV9) |
| | | 3.3 Program structure | • Peer-only vs. mixed sessions<br>• Flexible group composition<br>• Topic-dependent structure | "No, Not at all. I prefer only my peers" (ALHIV11) vs. "discussions that focus on the family... bring the adolescents and our parents together" (ALHIV1) |
| IMPLEMENTATION PROCESS | Theme 4: Participation Support Needs | 4.1 Material and logistical support | • Transportation assistance<br>• Reminder systems<br>• Financial support | "Sometimes by Sunday, you might be facing financial difficulties, so having support for transportation would be helpful" (Caregiver1) |
| | | 4.2 Diverse educational approaches | • Video content preferences<br>• Take-home materials<br>• Peer testimonials | "If there is a video of any of the topics you are sharing, you can show us" (Caregiver10) |
| | | 4.3 Digital alternatives | • Online meeting preferences<br>• Technology-mediated options<br>• Privacy through virtual | "Maybe you can create a group on the phone... every Sunday we can meet online" (ALHIV participant) |
| INTERVENTION CHARACTERISTICS | Theme 5: Content Preferences | 5.1 Comprehensive topics beyond HIV | • Emotional intelligence training<br>• Stigma coping strategies<br>• Mental health components | "We need training on emotional intelligence, how we cope with others, even those who don't know our condition" (ALHIV6) |
| | | 5.2 Interactive learning approaches | • Role-playing activities<br>• Drama demonstrations<br>• Experiential methods | "For us as caregivers, a short drama on social support or demonstrations of stress management would help us" (Caregiver3) |

represents a crucial enabling factor, as effective implementation requires properly trained facilitators who understand the intervention's techniques and principles. Health workers demonstrated an openness to professional development and a recognition that specialized training would be necessary to effectively deliver this psychosocial intervention.

*"I think personally, once you organize a workshop for us, it will be okay. And we, the staff, are readily available to attend any workshop that the group organizes."* [Health professional 2]

This quote clearly illustrates the health workers' positive disposition toward training opportunities. Their readiness for training represents a significant implementation advantage, as it addresses potential barriers related to staff resistance or lack of engagement.

**Table 5. Comparative synthesis of stakeholder perspectives on MFGT implementation enablers.**

| Enabler/ Theme | Caregivers | ALHIV | Health Workers | Consensus Level |
|---|---|---|---|---|
| Scheduling & Timing | Strong preference for Sunday after church; 14-week commitment acceptable; 2-hour sessions | Strong preference for Sunday after church; specific timing (1–4pm); concerns about session length | Flexibility in scheduling; aware of participant preferences; supportive of community timing | **High Consensus** |
| Location & Privacy | Comfort with hospital setting if privacy ensured; concerns about confidentiality | Mixed views on hospital vs. community; strong emphasis on privacy ("not an open place"); concerns about stigma | Hospital setting practical; awareness of privacy concerns; flexible to alternatives | **Moderate Consensus** |
| Program Structure | Value group learning; open to mixed sessions; prefer experienced facilitators including peers | Strong preference for peer-only sessions for certain topics; selective openness to family involvement | Support flexible composition; advocate for diverse facilitators including peer mentors | **Moderate-Low Consensus** |
| Material Support | Transportation assistance essential; reminder calls needed; acknowledge financial barriers | Transportation support important; reminder systems crucial; request for educational materials | Recognition of logistical barriers; supportive of providing material assistance | **High Consensus** |
| Educational Approaches | Prefer videos and pamphlets; value peer testimonials; appreciate experiential learning (drama) | Want diverse materials (videos, pamphlets); value peer experiences; prefer role-playing activities | Support multimodal approaches; recognize value of peer learning; endorse interactive methods | **High Consensus** |
| Digital Alternatives | Limited mention; less comfortable with technology-mediated options | Strong interest in online meetings; see technology as privacy solution; comfort with phone/virtual platforms | Recognize potential but aware of connectivity challenges; open to hybrid approaches | **Low Consensus** |
| Content Focus | HIV management important; interest in parenting skills; desire for family dynamics content | Comprehensive beyond HIV; emotional intelligence; stigma coping; mental health (including suicidal ideation) | Support holistic approach; recognize psychosocial needs; advocate for comprehensive curriculum | **Moderate-High Consensus** |
| Spiritual Integration | Strong desire for "word of God"; request for prayers to open/close sessions; faith as source of hope | Some mention of spiritual elements; less emphasis than caregivers | Respectful of community values; supportive of cultural/spiritual elements | **Moderate Consensus** |
| Facilitation Preferences | Want trained caregivers as co-facilitators; value experienced parent input | Prefer peer facilitators ("people of our age"); want mentors with lived experience | Advocate for multi-facilitator model; support training diverse leaders including peers | **Moderate Consensus** |
| Sustainability | Willingness to continue participation; see long-term value | Interest in ongoing support; value continuous learning | Strong commitment to continuation post-training; integration with existing programs | **High Consensus** |

• **High Consensus** (Green shading) - Strong agreement across all stakeholder groups.
• **Moderate Consensus** (Yellow shading) - General alignment with some nuanced differences.
• **Low Consensus** (Red shading) - Significant divergence requiring careful attention.

**Subtheme 2.2: commitment to program sustainability.** The FGD with the health workers revealed a strong commitment among them to ensure the long-term sustainability of the MFGT intervention beyond initial implementation. This dedication to program continuity represents a critical enabling factor, as sustainable interventions require local champions who are invested in maintaining program activities after external support ends.

*"Yes, there is a high possibility of continuing the intervention. You mentioned that you are going to train the staff. So, once you've trained us, we've gained that knowledge. If the program officially ends, we are the staff, we have acquired that knowledge. So, we'll continue with the intervention once the program has even ended."* (Health professional 1)

This quote provides powerful evidence of health workers' commitment to sustainability. The health workers emphasize knowledge transfer as a key sustainability mechanism through the repeated references to knowledge acquisition. This focus on knowledge transfer suggests health workers view the training not as a temporary exercise but as a permanent capacity-building opportunity that equips them with durable skills.

## CFIR domain: outer setting

### Theme 3: practical implementation considerations

**Subtheme 3.1: scheduling and Timing.** The discussions with caregivers and the adolescents revealed scheduling and timing as critical considerations for successful implementation of the MFGT program. Participants expressed clear preferences regarding when sessions should be held, highlighting the need to accommodate existing commitments and time constraints to maximize participation. The negotiation of appropriate timing emerged as a delicate balancing act between program requirements and participants' complex schedules, which include school, religious activities, and other community engagements.

*"We prefer Sunday after church"* (mentioned by multiple participants; caregivers and ALHIV)

This frequently expressed preference for Sunday sessions reflects the central role of religious activities in participants' weekly routines. The consistent mention of "after church" suggests that religious commitments take priority in participants' schedules, but also reveals a potential implementation opportunity—leveraging the time when participants are free from all activities. This timing preference demonstrates how the intervention must be accommodated within existing cultural and religious frameworks rather than competing with them.

The transcript further illustrates detailed timing considerations

*"When we come at 1pm… by 3-4pm we should be out of here"* (ALHIV 7)

This specific timeframe suggestion indicates participants have carefully considered when they could realistically attend sessions. The clearly defined start and end times (1:00–4:00 PM) suggest participants are weighing multiple commitments and need predictable scheduling to plan their day. The phrase "we are out of here" emphasizes the importance of adhering to the stated timeframe, suggesting that sessions running longer than expected could negatively impact attendance and retention.

Regarding program structure and commitment, the caregivers collectively agreed and echoed:

*"Once(1) in a week for fourteen(14) weeks. When we come we will spend 2 hours"* (Caregivers).

This quote demonstrates that caregivers have carefully thought about what kind of time commitment they can realistically manage. Their willingness to commit to a 14-week program suggests they recognize the potential benefits of the therapy despite their time limitations.

**Subtheme 3.2: location and environment.** The selection of an appropriate venue for the successful implementation of the MFGT emerged as a complex decision involving considerations of stigma, confidentiality, and practical accessibility. Participants' responses revealed tensions between different location options, with varying perspectives on the advantages and disadvantages of hospital versus community settings.

*"Yes I'm comfortable with a setting located within the Atua Government hospital or the HIV clinic"* (regarding hospital as meeting location) (ALHIV 5)

This straightforward affirmation from some participants indicates acceptance of the hospital as a potential meeting venue. For these individuals, the medical setting may provide a sense of legitimacy and appropriate context for discussions related to HIV management. The hospital represents a space already associated with their care, potentially reducing barriers to participation by integrating the intervention within established care routines.

However, the discussions also revealed significant concerns about privacy regardless of location

*"The environment we would meet shouldn't be an open place where other people would hear us"* (ALHIV 9)

The specific concern that "other people would hear us" highlights fears about inadvertent disclosure of HIV status through overheard conversations. This suggests that even when participants are comfortable with a particular venue, the specific characteristics of the meeting space within that venue remain critical considerations for implementation.

**Subtheme 3.3: program structure.** Participants provided detailed insights into desired group composition, facilitation approaches, and session formats that would enhance their comfort and engagement with the intervention. These structural preferences highlight the need for a nuanced, flexible approach to program implementation that accommodates different topics and participant dynamics.

Participants expressed clear views about group composition, indicating that the ideal structure would vary depending on the discussion topics and objectives of specific sessions.

*"I believe we should meet as a group because sharing individual experiences can help others learn from them."* (Caregiver 5)

This statement reflects an appreciation for the collective learning potential of group formats. The emphasis on shared experiences and mutual learning demonstrates that participants recognize the value of peer support and collective wisdom. This perspective aligns with core principles of MFGT, suggesting participant readiness for the intervention's foundational approach of learning through shared experiences.

However, discussions also revealed important nuances regarding group composition preferences. When the participant was asked whether they felt comfortable engaging in discussions that included their peers, parents, or siblings, one ALHIV responded hastily:

*No, Not at all. I prefer only my peers"* (ALHIV 11)

This clear preference for peer-only sessions highlights adolescents' need for safe spaces to discuss certain topics without parental presence. This preference has significant implications for program structure, suggesting the need for some sessions exclusively for adolescents.

Yet, participants also recognized value in mixed sessions for certain topics:

*"I think that for discussions that focus on the family and how to take care of an adolescent with HIV, you can bring the adolescents and our parents or caregivers together? "*(ALHIV 1)

The willingness to include family members for discussions about family dynamics and care suggests recognition that some challenges require collaborative approaches across the family system.

Regarding facilitation, participants expressed preferences for varied leadership:

*"Yeah!... I would like one of the caregivers/parents/guardians to be trained so they can lead program sessions some days"* (Caregiver 8)

*"Yes…. I would love people of our age or peers... who also have experiences about this encounter to facilitate the program"* (ALHIV 9)

*"It would be nice to have the health workers to lead and also sometimes a caregiver, because there are some with a lot of experience"* (Caregiver 13)

These statements collectively illustrate a desire for diverse facilitation approaches depending on the session content. This multi-facilitator preference suggests that implementation should include training for various types of session leaders rather than relying on a single facilitation model.

## CFIR domain: implementation process

### Theme 4: participation support needs

**Subtheme 4.1: material and logistical support.** The FGD with the caregivers and ALHIV highlighted practical material and logistical support as essential enablers for consistent participation in the MFGT program. Participants identified specific tangible resources and support mechanisms that would address barriers to attendance, emphasizing that practical considerations could significantly impact program engagement and retention.

Financial constraints emerged as a primary concern that could be addressed through targeted support mechanisms:

*"Sometimes by Sunday, you might be facing financial difficulties, so having support for transportation would be helpful."* (Caregiver 1)

This statement reveals the economic realities facing potential participants. Transportation support represents not merely a convenience but a necessity for ensuring consistent participation when financial constraints might otherwise force difficult choices between program attendance and other needs. The quote suggests that even short-term financial difficulties ("it will get to Sunday") could disrupt attendance patterns, highlighting the need for reliable transportation assistance throughout the program duration.

Participants also emphasized cognitive support through reminder systems

*"I feel that you should call us on Sunday morning. I do forget easily, so call us"* (Caregiver 4)

*"Please make it a point to call my sister just to remind her before Sunday. You can start calling her from Friday because I cannot come to the program alone without her"* (ALHIV 10)

These statements highlight the importance of structured communication systems to maintain engagement. The personal admission of forgetfulness ("I forget easily") demonstrates participants' self-awareness about potential barriers to attendance. The specific request for phone calls rather than other forms of communication suggests a preference for direct, personal reminders.

**Subtheme 4.2: preference for diverse educational approaches.** The FGD with caregivers and ALHIV revealed a strong desire for diverse educational tools and resources to enhance learning and retention within the MFGT program. Participants identified specific educational approaches that would support knowledge acquisition, reinforce key concepts, and facilitate behavior change beyond the immediate therapy sessions.

Participants expressed preferences for multimodal learning opportunities that engage different senses and learning styles:

*"If there is a video of any of the topics you are sharing, you can show us. Yes… I believe video contents are helpful learning tools"* (Caregiver 10)

This straightforward request for visual media demonstrates participants' desire for engaging, accessible educational content. Videos offer advantages of standardized information delivery while often being more memorable than verbal instruction alone. The suggestion indicates recognition that complex health information might be better understood and retained when presented visually.

Participants also valued take-home materials for ongoing reference:

*"Your team can also share with us reading materials like educational pamphlets so we can always make reference to the pamphlets in case we forget any of the things taught at the program"* (ALHIV 6)

This request demonstrates forward thinking about knowledge retention beyond the immediate session context. The desire for materials they can "always make reference to" indicates participants want to continue engaging with program content between sessions and potentially after the program concludes. This suggests readiness for self-directed learning and application of program concepts in daily life, which aligns with the goals of sustainable behavior change.

Perhaps most significantly, participants recognized the powerful educational impact of peer testimonials and role modeling:

*"I think parents with successful parenting styles or caregivers who have been able to successfully navigate the challenges of caring for an ALHIV can share their testimonies or stories with us. We can learn a lot from their experience."* (-Caregiver 5)

This profound statement reveals the transformative potential of peer experiences in changing perspectives and behaviors. This learning mechanism—observing peers who share similar challenges—appears to facilitate deep reflection and reconsideration of personal choices.

**Subtheme 4.3 digital alternatives.**  The FGD and in-depth interviews with the ALHIV revealed significant interest in digital communication platforms as potential alternatives or supplements to traditional in-person meetings for the MFGT program. This preference for digital options reflects participants' concerns about privacy, convenience, and accessibility, suggesting that technology-mediated communication could address multiple implementation barriers simultaneously.

Participants explicitly proposed virtual meeting formats as a solution to privacy concerns and logistical challenges:

*"Maybe you can create a group on the phone then you connect those that want to join, I mean those who will be joining this project. Then every Sunday we can meet online... There is no need for us to come there* (venue)." (ALHIV 2)

This suggestion demonstrates sophisticated thinking about how technology could transform program delivery. The emphasis that "there is no need for us to come there" highlights the perceived advantages of remote participation—eliminating travel requirements, reducing visibility concerns, and potentially increasing accessibility.

## CFIR domain: intervention characteristics

### Theme 5: content preferences

**Subtheme 5.1: comprehensive topics beyond HIV.**  The FGDs revealed participants' desires for a holistic approach to the MFGT program that addresses not only HIV management but also broader psychosocial needs. Their content preferences demonstrate sophisticated understanding of the interconnected nature of physical health, emotional wellbeing, and social functioning.

Participants articulated specific topic areas they wanted covered, showing insights into the diverse challenges they face:

*"We need training on emotional intelligence, how we cope with others, even those who don't know our condition, how we cope with them."* (ALHIV 6)_

The reference to "emotional intelligence" shows understanding that managing one's emotional responses is a critical skill for wellbeing. Particularly notable is the phrase "how we cope with others, even those who don't know our condition," which highlights the daily social navigation required—maintaining relationships with people who may not know their status while managing the stress of potential disclosure or discovery.

Participants also emphasized the importance of addressing mental health components:

"*We can discuss about our perceptions and the anxieties we have sometimes. For instance, some people with this condition have a perception that they will die soon, yes. So, you will have to talk more about the suicidal thoughts we have.*" (ALHIV 1)

These requests demonstrate awareness of the psychological impact of HIV. The reference to perceptions about mortality reveals awareness of how cognitive patterns (thinking they will "die soon") impact psychological wellbeing. This sophisticated understanding of the thought-emotion connection indicates readiness for cognitive-behavioral approaches typically included in therapeutic interventions. The finding also underscores the urgency of implementing MFGT in this setting, highlighting that such programs should be considered potentially life-saving interventions addressing severe mental health crises, including suicidal thoughts, among ALHIV

Participants articulated the need for training on how to cope/face stigma:

"*We (the ALHIV) need to be trained on how to face people, gossip and how to overcome stigma*" (ALHIV 4)

Stigma was mentioned as a pervasive concern that significantly impacts the lives of adolescents living with HIV and their families. Participants identified specific stigma-related challenges and expressed strong desires for the MFGT program to address strategies for managing these difficult social situations.

**Subtheme 5.2: interactive learning approaches.** Participants advocated for experiential approaches that would enhance engagement, retention, and practical application of therapeutic concepts beyond the session environment. Both caregivers and ALHIV expressed enthusiasm for learning methods that transcend traditional didactic instruction:

"*For us as caregivers, a short drama on social support or demonstrations of stress management would help us to retain and practice what we are taught*" (Caregiver 3)

This statement from a caregiver demonstrates sophisticated understanding of adult learning principles. The specific request for "short drama" and "demonstrations" indicates recognition that observational learning can be more effective than verbal instruction alone.

Similarly, adolescents emphasized the value of experiential learning:

"*I think role-playing how we can give support to our parents and siblings and also how we can receive support would help us*"(ALHIV 5)

This adolescent's request for role-playing activities demonstrates understanding that interpersonal skills are best developed through practice rather than discussion alone. These requests reveal sophisticated understanding of effective learning methods. The mention of "drama" and "role playing" indicates preference for experiential learning approaches that facilitate skill development through practice rather than merely gaining theoretical knowledge.

**Subtheme 5.3 spiritual support.** The focus group discussions revealed spirituality as an important dimension that participants wanted integrated into the MFGT program. This emphasis on spiritual elements reflects the cultural context in which the intervention will be implemented and highlights the role of faith as a coping mechanism and source of strength for many participants.

Participants explicitly requested the inclusion of religious content as a core program component:

"*Let's add the word of God because it would help us get hope*" (Caregiver 9)

This statement powerfully illustrates the perceived therapeutic value of spiritual content. The direct connection between "the word of God" and "hope" demonstrates how religious teachings are viewed as sources of emotional sustenance and psychological resilience.

Participants also expressed desires for incorporating spiritual practices into the program structure:

"*We can also begin and close each training session with prayer*" (Caregiver 4)

This suggestion for framing sessions with prayer reflects the importance of spiritual rituals in creating sacred or meaningful spaces.

## Discussion

This study identified six key enablers for implementing Multiple Family Group Therapy (MFGT) among adolescents living with HIV in Ghana's Lower Manya Krobo District: existing support systems and infrastructure, healthcare worker willingness to participate, practical implementation considerations, participation support needs, content preferences for comprehensive topics, and spiritual support integration. Triangulation across stakeholder groups revealed consensus on privacy needs, scheduling preferences, and spiritual integration, while showing divergences in facilitation preferences and technology adoption.

Mapping findings to CFIR's five domains demonstrates implementation science relevance. Inner Setting enablers included existing support systems (Community Refill Program, adolescent clinic days) and organizational readiness. Outer Setting emphasized community scheduling (Sunday after church) and cultural integration (spiritual elements). Individual Characteristics highlighted healthcare worker training readiness and sustainability commitment. Intervention Characteristics indicated necessary adaptations: comprehensive content beyond HIV, interactive approaches, and flexible structures (peer-only versus mixed). Implementation Process identified material support needs (transportation, reminders), diverse educational approaches (videos, role-playing, testimonials), and vulnerable population strategies. Successful MFGT implementation in resource-limited settings requires coordinated attention across all domains, particularly cultural adaptation (Outer Setting) and sustainability planning (Individual Characteristics and Implementation Process).

These findings align with several studies on MFGT implementation while revealing unique considerations for resource-limited settings. The importance of existing support infrastructure mirrors findings from previous studies [6,35], which demonstrated that successful MFGT programs typically build upon established healthcare structures rather than creating entirely new systems. This alignment suggests that Ghana's healthcare system has foundational elements necessary for MFGT integration, challenging assumptions that resource constraints automatically preclude complex psychosocial interventions. Similarly, our finding regarding healthcare worker commitment aligns with Damschroder et al.'s [36] CFIR framework, which identifies provider readiness as a critical implementation determinant. The healthcare worker dedication identified in our study suggests that human resource availability may compensate for material resource limitations, indicating that implementation strategies should prioritize capacity building over infrastructure development.

However, our study reveals contextual factors not prominently featured in high-income country literature, suggesting that successful MFGT implementation in sub-Saharan Africa requires a fundamentally different approach than direct replication of Western models. The unanimous preference for Sunday scheduling after religious activities contrasts sharply with Western studies where scheduling flexibility typically focuses on work-life balance rather than religious observance. Peterson et al. [37] noted similar cultural adaptation needs in their review of intervention modifications while Mohr, Riper and Schueller [38] found that implementation designs that accommodate participant schedules and preferences show 2–3 times better retention rates than rigid implementations.

However, our findings provide specific guidance for sub-Saharan African contexts. The convergence of stakeholder preferences around cultural elements implies that cultural adaptation is not optional but essential for implementation

success. This contrasts with adaptation approaches that treat cultural modifications as supplementary enhancements rather than core requirements.

The emphasis on spiritual integration in our study differs markedly from secular approaches common in Western MFGT implementations and represents a critical finding for implementation in similar contexts. While Kpobi and Swartz [21] documented the importance of spirituality in Ghanaian mental health conceptualizations, our study is among the first to demonstrate how this translates into specific implementation requirements for family-based HIV interventions. The participants' emphasis on incorporating "the word of God" and prayer into sessions highlights how spiritual elements serve specific therapeutic functions—particularly providing hope and meaning—that complement clinical approaches [21]. This finding suggests that successful MFGT implementation in similar contexts requires genuine integration of spiritual practices rather than superficial cultural adaptations. The convergence of these preferences across stakeholder groups indicates that experiential and spiritual elements may be as critical to intervention success as evidence-based therapeutic techniques, pointing toward hybrid models that combine the strengths of both Western clinical approaches and indigenous healing practices.

Our finding regarding transportation support as a participation enabler aligns with Ingoldsby [39] study, which found that addressing practical barriers significantly improved engagement in group-based interventions for people living with HIV. However, our study extends this by identifying specific support mechanisms including reminder systems and material resources that facilitate sustained participation. These practical considerations suggest that resource-limited settings may have unique advantages for MFGT implementation, particularly strong community support systems and healthcare worker dedication that can facilitate intervention sustainability. However, these advantages can only be leveraged through implementation approaches that recognize and build upon existing cultural and organizational strengths.

The preference for digital alternatives among adolescents, despite connectivity challenges in rural Ghana, challenges assumptions about technology adoption in resource-limited settings and suggests that technology-mediated delivery could address privacy concerns while potentially expanding reach. This finding indicates that even basic mobile technologies could enhance intervention delivery and that privacy concerns in stigmatized health conditions require innovative solutions that go beyond traditional confidentiality measures [4,40]. The preference for digital alternatives and specific scheduling requests suggest that implementation success may depend on creative approaches to reducing visibility and social risk associated with program participation.

The divergent preferences across stakeholder groups regarding facilitation approaches and technology comfort suggest that flexible implementation models accommodating different needs within the same program may be more effective than standardized approaches. Healthcare professionals emphasized sustainability benefits of training diverse facilitators and integrating MFGT into existing clinical workflows, while adolescents showed stronger preferences for peer-led sessions and digital alternatives. Caregivers occupied a middle position, appreciating both professional guidance and peer testimonials but showing less comfort with technology-mediated interventions. These differences indicate that MFGT implementation in similar contexts should prioritize cultural responsiveness, leverage existing support systems, and maintain flexibility to accommodate diverse stakeholder needs while ensuring robust privacy protection mechanisms. Rather than viewing these divergences as implementation challenges, they suggest opportunities for adaptive program design that can simultaneously meet different stakeholder priorities while maintaining core therapeutic principles.

The strong preference for experiential learning methods and spiritual integration reveals an important gap between conventional therapeutic approaches and participant expectations [41]. Participants' explicit requests for role-playing, drama, and religious content suggest that purely didactic, secular interventions may miss key engagement opportunities in this cultural context. Rather than viewing these preferences as accommodations to standard protocols, they may represent essential therapeutic components that enhance intervention effectiveness.

## Assessing the readiness of the Ghanaian system

The Ghanaian health system in Lower Manya Krobo District demonstrates considerable readiness for MFGT implementation, with key strengths including established HIV support programs, dedicated adolescent clinic days, committed health workers showing readiness for training, potential for cultural integration through local festivals, and multi-stakeholder engagement in the planning process. These elements align with implementation science research identifying infrastructure integration, provider readiness, and cultural adaptation as critical determinants of success, suggesting a promising foundation for program implementation.

## Recommendations for implementation

Based on the study findings and existing evidence, implementing MFGT in the Lower Manya Krobo District should follow a prioritized, comprehensive approach that aligns with both stakeholder consensus and practical feasibility. Highest-priority actions include cultural integration—such as aligning sessions with community rhythms (post-church Sundays) and incorporating spiritual elements—and leveraging existing adolescent clinic structures, as these are universally supported and readily implementable.

High-priority strategies involve training diverse facilitators, including healthcare workers, experienced caregivers, and peer mentors, as well as providing practical support through transportation assistance and reminder systems. These actions are strongly supported by stakeholders and feasible with moderate resource investment. Moderate-priority recommendations include designing flexible program options that accommodate both peer-only and family sessions and integrating digital platforms to address privacy concerns; while these approaches are preferred by some participants, they present implementation challenges due to technology access and caregiver comfort levels.

The approach should also include establishing a cultural adaptation workgroup to ensure content resonates with local values, implementing privacy protections such as anonymous participation options, designing evaluation measures for both health and implementation outcomes, and creating a phased sustainability plan that gradually transfers ownership to local health workers while maintaining supervision. This multifaceted, prioritized strategy addresses identified enablers and barriers while building on existing strengths in the Ghanaian health system, maximizing the potential for successful, culturally responsive, and sustainable MFGT implementation.

## Limitations of the study

Despite providing valuable insights into MFGT implementation enablers for adolescents living with HIV in Ghana's Lower Manya Krobo District, this study has several limitations. The qualitative design limits generalizability to other settings with different resource levels, cultural contexts, and HIV prevalence rates. Our findings represent a snapshot of perspectives before actual implementation, and participants' perceptions may change once they experience the intervention firsthand. The study did not capture the viewpoints of other important stakeholders such as community leaders, religious figures, school officials, and policy makers, potentially limiting our understanding of community-level enablers. Additionally, we focused primarily on enablers rather than barriers, and did not quantitatively assess the relative importance of different enablers, making it difficult to prioritize implementation strategies. Selection bias may have favored participants already engaged with HIV services, while social desirability bias may have influenced responses, particularly given the hierarchical nature of healthcare relationships in the Ghanaian context. The research team's positionality likely influenced data collection and interpretation. The first author's academic status and institutional affiliation may have created power differentials, leading participants—particularly ALHIV and caregivers—to provide socially desirable or "academically acceptable" responses. Language translation and the dual role of the research assistant as interpreter and note-taker could have further affected responses. Additionally, conducting data collection within healthcare facilities may have blurred boundaries between research and clinical care, influencing both participants' and healthcare workers' responses. Overall, these

factors suggest that while the study captured key enablers for MFGT implementation, some responses may reflect socially desirable answers rather than fully unfiltered stakeholder perspectives. Nevertheless, this study provides important foundational knowledge for implementing MFGT in resource-limited settings and highlights the value of stakeholder engagement in adaptation and implementation planning.

## Conclusion

The findings from this study reveal significant enablers for MFGT implementation in the Lower Manya Krobo District, many of which align with established evidence on successful implementation factors. The Ghanaian health system shows meaningful readiness through existing infrastructure, provider willingness, and cultural support mechanisms. However, addressing privacy concerns, resource constraints, and cultural adaptation needs will be critical for successful implementation. By leveraging the identified enablers and systematically addressing potential barriers, MFGT implementation in this context has strong potential to improve outcomes for adolescents living with HIV and their families. The recommendations provided offer a roadmap for culturally responsive, sustainable implementation that builds on existing strengths while addressing identified challenges.

## Supporting information

**S1 Text. Guide for FGD and interview for ALHIV.**
(DOCX)

**S2 Text. FGD guide for caregivers.**
(DOCX)

**S3 Text. FGD guide for healthcare workers.**
(DOCX)

**S1 File. COREQ. checklist.**
(DOCX)

**S1 Data. Transcript.**
(DOCX)

**S2 Data. Transcript.**
(DOCX)

**S3 Data. Transcript.**
(DOCX)

## Author contributions

**Conceptualization:** Dorothy Serwaa Boakye, Samuel Adjorlolo.

**Formal analysis:** Dorothy Serwaa Boakye, Samuel Adjorlolo.

**Investigation:** Dorothy Serwaa Boakye.

**Methodology:** Dorothy Serwaa Boakye, Samuel Adjorlolo.

**Project administration:** Dorothy Serwaa Boakye.

**Supervision:** Samuel Adjorlolo.

**Writing – original draft:** Dorothy Serwaa Boakye, Samuel Adjorlolo.

**Writing – review & editing:** Dorothy Serwaa Boakye, Samuel Adjorlolo.

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
