## [Decision Letter · Decision Letter 0]

13 Aug 2025

PGPH-D-25-01360

Potential Enablers for the implementation of multiple family group therapy intervention in the Lower Manya Krobo District, Ghana: Perspectives of multiple stakeholders.

Dear Dr. Adjorlolo,

Thank you for submitting your manuscript to PLOS Global Public Health. After careful consideration, we feel that it has merit but does not fully meet PLOS Global Public Health’s publication criteria as it currently stands. Therefore, we invite you to submit a revised version of the manuscript that addresses the points raised during the review process.

We look forward to receiving your revised manuscript.

Kind regards,

Prof Razak Gyasi, PhD, PD

Academic Editor

Journal Requirements:

1. Please amend your online detailed Financial Disclosure statement. This is published with the article. It must therefore be completed in full sentences and contain the exact wording you wish to be published.

a) State the initials, alongside each funding source, of each author to receive each grant, if applicable. For example: "This work was supported by the National Institutes of Health (####### to AM; ###### to CJ) and the National Science Foundation (###### to AM)."

For more information, please go to our submission guidelines:

https://journals.plos.org/globalpublichealth/s/submission-guidelines#loc-financial-disclosure-statement

2. Please ensure that the funders and grant numbers match between the Financial Disclosure field and the Funding Information tab in your submission form. Note that the funders must be provided in the same order in both places as well.

3. Please send a completed ‘Competing Interests’ statement, including any COIs declared by your co-authors. If you have no competing interests to declare, please state “The authors have declared that no competing interests exist”. Otherwise please declare all competing interests beginning with the statement ‘I have read the journal's policy and the authors of this manuscript have the following competing interests:’

For more information, please go to our submission guidelines:

https://journals.plos.org/globalpublichealth/s/submission-guidelines#loc-competing-interests

4. In the online submission form, you indicated that “Information regarding this paper is available upon reasonable request from the corresponding author S.A.”.

a) In a public repository,

b) Within the manuscript itself, or

c) Uploaded as supplementary information.

5. Please ensure that you refer to Table 1 in your text as, if accepted, production will need this reference to link the reader to the table.

6. We have noticed that you have uploaded Supporting Information files, but you have not included a list of legends. Please add a full list of legends for your Supporting Information files before or after the references list.

Additional Editor Comments (if provided):

Reviewers' comments:

Reviewer's Responses to Questions

**Comments to the Author**

1. Does this manuscript meet PLOS Global Public Health’s publication criteria?

Reviewer #1: Partly

Reviewer #2: Yes

Reviewer #3: Yes

Reviewer #4: Yes

2. Has the statistical analysis been performed appropriately and rigorously?

Reviewer #1: Yes

Reviewer #2: No

Reviewer #3: Yes

Reviewer #4: N/A

3. Have the authors made all data underlying the findings in their manuscript fully available (please refer to the Data Availability Statement at the start of the manuscript PDF file)?

Reviewer #1: Yes

Reviewer #2: Yes

Reviewer #3: Yes

Reviewer #4: Yes

4. Is the manuscript presented in an intelligible fashion and written in standard English?

Reviewer #1: Yes

Reviewer #2: Yes

Reviewer #3: Yes

Reviewer #4: Yes

Reviewer #1: Recommendation: Minor Revision

1. Overall Assessment

This is a culturally appropriate and methodologically robust qualitative study that identifies enabling contexts for MFGT implementation among adolescents living with HIV (ALHIV) in a resource-limited environment. The study is rich in perspectives from participants, primarily through stakeholder triangulation involving caregivers, ALHIV, and health workers. The study presents insightful additions to implementation science in global mental health and HIV treatment, particularly in sub-Saharan Africa.

2. Major Strengths

Relevance & Originality: Emphasizes a previously unexplored yet critical area, such as mental health care provision for ALHIV in SSA, through a family-based model.

Methodologically Rigorous Qualitative Design: Thematic analysis, use of triangulation, and reflexivity are outstanding.

Stakeholder Involvement: Incorporation of several perspectives (youth, care providers, caregivers) demonstrates ethical, equitable research

Policy Value: Provides in-depth, practical information on program feasibility, adaptation, and cultural considerations.

3. Major Comments and Recommendations

B. Data Availability and Transparency

Issue: The Data Availability Statement is limiting data to "reasonable request," which is not in alignment with PLOS's open data policy.

Recommendation: Authors should specify what data (e.g., anonymised transcripts, coding schemes) are available to be shared and post them to a public repository or offer a justified exemption (e.g., participant privacy issues).

B. Analytical Depth and Triangulation

Issue: Although the methodology mentions triangulation, a more explicit comparative synthesis is needed across stakeholder groups in the findings.

Recommendation: Include a paragraph in the discussion findings mentioning points of consensus or divergence among all three stakeholder groups.

C. Limitations and Reflexivity

Issue: The limitations discussion could provide a more precise explanation of how the researcher's positionality may have influenced data interpretation, particularly in light of power differentials.

Recommendation: Discuss further reflexivity to explain how training of the research team or institutional status may have affected interaction with vulnerable respondents.

D. Conceptual Framing

Issue: The CFIR is mentioned in tool development, but not meaningfully integrated into the analysis or discussion.

Recommendation: Briefly map the five CFIR domains (intervention characteristics, outer setting, inner setting, individuals, and implementation process) to the identified themes to enhance theoretical clarity.

4. Minor Comments

Describe what is meant by "Form 4" in Table 1, caregiver education.

Some sentences are too long and could be condensed; a slight edit to improve clarity and flow is preferable. Ensure consistency in referencing styles; some in-text citations use the author-date format, while others use the numeric format.

Reviewer #2: Abstract

"There are notable enablers of the MFGT for ALHIV in Lower Manya Krobo District.

Researchers and practitioners are advised to pay critical attention to enablers in their quest to

implement MFGT to address the mental health burden of ALHIV".

This conclusion does not summary the findings, Researchers and practitioners are advised to pay................ is not an actionable recommendation. Kindly provide an actionable recommendation

Keywords: Multiple Family Group Therapy; implementation science; adolescents living with

HIV; perspectives; multiple stakeholders

these are not keyword, they are too long

Methodology

1. This study lacks Theoretical issues

2. Study population cannot be seen in the study and exclusion is also missing

3. Sample sizes were determined based on achieving theoretical saturation while remaining

logistically feasible. The final sample included 12 ALHIV, 13 caregivers, and 5 healthcare

professionals, for a total of 30 participants. This multi-stakeholder approach allowed for

triangulation of perspectives and comprehensive understanding of implementation enablers from

different vantage points within the healthcare ecosystem.......... This is not well justified

4. Research team is also missing, who were the research team. Do all team members had prior experience with qualitative interviews?. training session ensured consistency in the use of research instruments and interview techniques is also missing in the work. The recruitment period for this study started is also missing in the work

5.Data collection occurred through two distinct modalities.

6. "Individual interviews were conducted in English by the 1st author using remote methods, either via

telephone or Google Meet videoconferencing platform" what if there was network issues how did you address such and why online interview do you have any justification for it?.

7. FGDs lasted approximately 90-120 minutes, while interviews lasted 60-90 minutes. You mean the interview last more than an hour?.

8. Challenges encountered. What was the challenges encounter in this study

9. Trustworthiness and rigor. You cannot write a full write on Trustworthiness and rigor without citing a study or without using the Consolidated Criteria for Reporting Qualitative Research (COREQ)

10. this is not a proper way of writing the trustworthiness and rigor "Rigor and Reflexivity

Several strategies were employed to ensure trustworthiness of the findings. Credibility was

enhanced through triangulation of data sources (multiple stakeholder groups) and methods (FGDs

and interviews), member checking (sharing preliminary findings with selected participants for

feedback), and peer debriefing among the research team. Dependability was established through

maintenance of an audit trail documenting all methodological decisions and analytical processes.

Confirmability was addressed through researcher triangulation in the analysis process and

reflexive journaling to acknowledge and mitigate potential researcher biases.

Reflexivity was maintained throughout the research process through regular team discussions

about how researchers' backgrounds, assumptions, and positions might influence data collection

and interpretation. The research team included a professor and PhD candidate, both are researchers

with diverse professional backgrounds and expertise in qualitative research, allowing for multiple

perspectives in the interpretation of findings. Particular attention was paid to power dynamics

between researchers and participants, especially with vulnerable populations such as ALHIV".

kindly use this study to restructure your rigor writing https://pmc.ncbi.nlm.nih.gov/articles/PMC12224659/

11. so what were your Study variables? you dont have study variables why?

Result

1. Table 1: Sociodemographic characteristics of participants. Table 1 is AI generated please can justify why you are using AI generated table. The table is even scattered. the table should be simple. eg. Variables - Frequency - Percentage rather than the AI generated table

2. Thematic Result

The presentation of the result is not appropriate. Where is your thematic table. The thematic table should consist of the main theme, sub theme and codes before interpretation of the results with each sub theme. How many sub theme and codes were generated ?. All these needs to appear in the work. Kindly see this qualitative work from plos mental health to restructure your result https://journals.plos.org/mentalhealth/article?id=10.1371/journal.pmen.0000310 and you can also use this link as well https://pmc.ncbi.nlm.nih.gov/articles/PMC12224659/

3. Re structure all the result section again

Discussion

The discussion is not good. You need write a summary of the results then u compare it to other studies then you tell us what implies and what your study suggest. it is not about discussing other literatures. Please kindly re-write your discussion again

Reviewer #3: Your paper demonstrates sound methodological and ethical rigor, presents conclusions appropriately supported by the data, and makes a significant contribution to global public health by addressing the implementation of a psychosocial intervention for adolescents living with HIV (ALHIV) in a resource-limited setting.

The recommendations for implementation (hybrid model, diverse facilitators, cultural adaptations) are comprehensive but presented without clear prioritization. To enhance practical utility, you can consider briefly indicating which recommendations are most critical or feasible based on the data. Also expand the discussion to explicitly address how findings could inform implementation in similar SSA settings.

Reviewer #4: The author did quite well in identifying the enablers for MFGT implementation in the Lower Manya Krobo District, Ghana. The following points are my suggestions regarding this work:

In the first paragraph of the introduction, the statement: Previous studies has... should read: Previous studies have...

Table 1 is too long; it is suggested to be divided into three parts. Remove the column with the heading 'Population Group', which contains repeated names of subpopulation groups. Each table should have a heading, for example:

Table 1: Sociodemographic Characteristics of Adolescents Living with HIV

Table 2: Sociodemographic Characteristics of Healthcare Workers

Table 3: Sociodemographic Characteristics of Caregivers

On page 21, the participant ALHIV 1 articulated the statement: "So, you will have to talk more about the suicidal thoughts we have." In the explanation that follows this statement, the author should add the need to address suicidal thoughts being experienced by the ALHIV

Page 22: The statement, "I think we role-playing, how we can give support to our parents and siblings, and how we can receive support would help us," is not properly phrased, especially the first part of the statement. The author should review the way the statement is presented for more clarity.

**Do you want your identity to be public for this peer review?** For information about this choice, including consent withdrawal, please see our Privacy Policy

Reviewer #1: **Yes:** Abimbola Adegoke

Reviewer #2: No

Reviewer #3: No

Reviewer #4: **Yes:** Taiwo Olufemi Abiona

---

## [Decision Letter · Decision Letter 1]

8 Oct 2025

PGPH-D-25-01360R1

Potential Enablers for the implementation of multiple family group therapy intervention in the Lower Manya Krobo District, Ghana: Perspectives of multiple stakeholders.

Dear Dr. Adjorlolo,

Thank you for submitting your manuscript to PLOS Global Public Health. After careful consideration, we feel that it has merit but does not fully meet PLOS Global Public Health’s publication criteria as it currently stands. Therefore, we invite you to submit a revised version of the manuscript that addresses the points raised during the review process.

We look forward to receiving your revised manuscript.

Kind regards,

Prof Razak Gyasi, PhD, PD

Academic Editor

Journal Requirements:

Additional Editor Comments (if provided):

Reviewers' comments:

Reviewer's Responses to Questions

**Comments to the Author**

Reviewer #1: (No Response)

Reviewer #4: All comments have been addressed

publication criteria?

Reviewer #1: Yes

Reviewer #4: Yes

3. Has the statistical analysis been performed appropriately and rigorously?

Reviewer #1: Yes

Reviewer #4: Yes

4. Have the authors made all data underlying the findings in their manuscript fully available (please refer to the Data Availability Statement at the start of the manuscript PDF file)?

Reviewer #1: Yes

Reviewer #4: Yes

5. Is the manuscript presented in an intelligible fashion and written in standard English?

Reviewer #1: Yes

Reviewer #4: Yes

Reviewer #1: This revised paper is a fascinating contribution to the literature on family-centered mental health interventions among adolescents living with HIV (ALHIV) from sub-Saharan Africa. That you managed to integrate the perspectives of numerous stakeholders and culture-bound strategies for determining feasibility earns accolades.

Below are the detailed comments for further refinement:

A. Triangulation and Comparative Synthesis:

You increased the integration of viewpoints within the different groups of stakeholders. Additional synthesis within the text clarifies the argument. A brief comparative table summarizing the extent of consensus and disagreement among caregivers, ALHIV, and health workers would facilitate the reader's use.

B. Reflexivity and Positionality:

Reflexivity was slightly widened. It would be desirable, though, to describe briefly the degree to which the positionality of the researchers (e.g., professional experience, institutional affiliation, cultural familiarity with participants’ culture) affected interaction with the participants or the interpretation of the answers—particularly considering the power imbalance involved when working with vulnerable groups of adolescents.

C. CFIR Integration:

The reference to CFIR on your tool development is helpful. To enhance theoretical clarity, we would like you to succinctly map your results to the five CFIR domains in a summary paragraph or a very short table within the Discussion section. Doing so would make the study more relevant for implementation science experts.

D. Ethical Considerations and Data Transparency:

Include a statement ensuring IRB/ethics approval was granted or waived, and by what institution.

Discuss the way participant consent and confidentiality were managed, particularly for adolescents.

The current Data Availability Statement doesn't quite align with PLOS policy. If you're unable to make all your data publicly available, please explain (e.g., due to the terms of ethical approval or participant anonymity). Alternatively, deposit a de-identified codebook or summary data into a repository.

E. Minor Editing Suggestions:

Ensure consistency of referencing style (author-date or numerical).

Define such items as the table entry for "Form 4" for foreign readers, which you've just done,

concise sentences where possible for greater clarity.

Check the usage of acronyms and spell them out on first occurrence.

Conclusion:

This is a considerate and timely piece of work. The few minor areas for revision can be easily addressed, and when clarified, will form an impressive addition to the literature on youth mental health services, family-centered programs, and culturally competent program delivery under constrained resources.

Reviewer #4: I have compared the previously submitted write-up with this current one, the authors have adequately addressed the comments I raised in a previous round of review.

**Do you want your identity to be public for this peer review?** For information about this choice, including consent withdrawal, please see our Privacy Policy

Reviewer #1: **Yes:** Abimbola Adegoke

Reviewer #4: **Yes:** Taiwo Olufemi Abiona

---

## [Editor Report · Decision Letter 2]

20 Nov 2025

Potential Enablers for the implementation of multiple family group therapy intervention in the Lower Manya Krobo District, Ghana: Perspectives of multiple stakeholders.

PGPH-D-25-01360R2

Dear Dr. Adjorlolo,

We are pleased to inform you that your manuscript 'Potential Enablers for the implementation of multiple family group therapy intervention in the Lower Manya Krobo District, Ghana: Perspectives of multiple stakeholders.' has been provisionally accepted for publication in PLOS Global Public Health.

Best regards,

Professor Razak M Gyasi, PhD, PD

Academic Editor